# Differential Action of Connexin Hemichannel and Pannexin Channel Therapeutics for Potential Treatment of Retinal Diseases

**DOI:** 10.3390/ijms22041755

**Published:** 2021-02-10

**Authors:** Mohd N. Mat Nor, Ilva D. Rupenthal, Colin R. Green, Monica L. Acosta

**Affiliations:** 1School of Optometry and Vision Science, University of Auckland, Auckland 1142, New Zealand; nasirmnor@unisza.edu.my; 2Faculty of Medicine, Universiti Sultan Zainal Abidin, Kuala Terengganu 20400, Malaysia; 3Buchanan Ocular Therapeutics Unit, Department of Ophthalmology and New Zealand National Eye Centre, University of Auckland, Auckland 1142, New Zealand; i.rupenthal@auckland.ac.nz; 4Department of Ophthalmology and New Zealand National Eye Centre, University of Auckland, Auckland 1142, New Zealand; c.green@auckland.ac.nz; 5School of Optometry and Vision Science, New Zealand National Eye Centre, Centre for Brain Research, Brain Research New Zealand—Rangahau Roro Aotearoa, University of Auckland, Auckland 1142, New Zealand

**Keywords:** retina, light damage, connexin43, pannexin, tonabersat, probenecid, inflammation, electroretinogram

## Abstract

Dysregulation of retinal function in the early stages of light-induced retinal degeneration involves pannexins and connexins. These two types of proteins may contribute to channels that release ATP, leading to activation of the inflammasome pathway, spread of inflammation and retinal dysfunction. However, the effect of pannexin channel block alone or block of both pannexin channels and connexin hemichannels in parallel on retinal activity in vivo is unknown. In this study, the pannexin channel blocker probenecid and the connexin hemichannel blocker tonabersat were used in the light-damaged rat retina. Retinal function was evaluated using electroretinography (ERG), retinal structure was analyzed using optical coherence tomography (OCT) imaging and the tissue response to light-induced injury was assessed immunohistochemically with antibodies against glial fibrillary acidic protein (GFAP), Ionized calcium binding adaptor molecule 1 (Iba-1) and Connexin43 (Cx43). Probenecid did not further enhance the therapeutic effect of connexin hemichannel block in this model, but on its own improved activity of certain inner retina neurons. The therapeutic benefit of blocking connexin hemichannels was further evaluated by comparing these data against results from our previously published studies that also used the light-damaged rat retina model. The analysis showed that treatment with tonabersat alone was better than probenecid alone at restoring retinal function in the light-damaged retina model. The results assist in the interpretation of the differential action of connexin hemichannel and pannexin channel therapeutics for potential treatment of retinal diseases.

## 1. Introduction

Tissue inflammation involves cellular communication and the roles of connexin channels in this process have been elucidated using a variety of experimental tools and animal models [1,2]. Two large membrane pores have been proposed to play a role in response to injury or disease—the connexin hemichannel and the pannexin channel. A connexin hemichannel (or connexon) is comprised of six connexin subunits, with two connexons from adjacent cells docking together to form a gap junction channel linking the cytoplasm of the two cells. Most organs express different types of connexins and form heterotypic or homotypic gap junction channels. Connexin43 (Cx43) is expressed in various cell types and contributes to normal physiology. In contrast to the gap junction, opening of Cx43 undocked hemichannels implicates injury, extracellular cell signaling and a mechanical stimulation response in various tissues [3,4]. Prior to docking, hemichannels need to remain primarily closed or they form a large, non-specific membrane pore exposing the cell cytoplasm to the extracellular milieu [5]. The opening of connexin hemichannels is the result of molecular and physical insults that disrupt normal homeostasis [4,5,6].

Pannexins share sequence homology with the gap junction proteins of invertebrates (innexins) and have a similar topological structure to connexins, but do not share sequence homology with connexins. In the body, pannexin channels are vital for cell homeostasis and long-term blockage of these pores causes dysregulated tissue function [7]. In most cases, pannexins do not form cell-cell communication channels, but are instead primarily tightly regulated membrane channels [8,9]. In the eye, Cx43 is expressed in the lens and in the glia of the retina, blood vessels and epithelial cells [10]. The distribution of connexin isoforms expressed in the retina across species is listed in detail in Table 1 of Danesh-Meyer et al. [4]. Pannexin1 (Panx1) is expressed in the anterior part of the eye but in the retina Panx1 and Panx2 are widely distributed including ganglion cells and various cells of the inner retina [11]. A role for pannexin channels in the homeostasis of the retina has been suggested given their wide distribution in the eye [11].

In the light-damaged rat retina, a model of retinal degeneration, hemichannels formed by Cx43 in the choroid, retinal pigment epithelium (RPE) and retina significantly contribute to the inflammatory response [12,13,14,15,16]. It is now understood that a substantial component of inflammation is mediated by the opening of unopposed Cx43 hemichannels [2,15,17,18], involved in the initiation stages and progression of the inflammatory reaction. However, the role of pannexins in the light-damaged retina is unknown. Pannexin channels function release ATP and metabolites for cellular communication and for paracrine signaling [19,20,21,22,23]. Pannexin proteins are also upregulated in tissues with chronic inflammation [24,25], and have been proposed to enhance ATP release in the retinal and choroidal environment leading to cell death by apoptotic mechanisms [21,22,26]. Pannexin channels have been suggested as the source of retinal damage in other retinal inflammation conditions. For example, in an in vitro ischemia/reperfusion model, about a third of ATP release during ischemia was found to be pannexin-channel mediated while two thirds was connexin hemichannel-mediated, although during the reperfusion phase only connexin hemichannels were responsible for ATP release [15]. Connexin hemichannels and pannexin channels have also been linked to the pathophysiology of ischemia in other organs, including the occurrence of seizures and in diseases of the eye [4,27,28]. Given the vital role of ATP in activation of the inflammasome [29] and the mixture of evidence about the roles of both pannexin channel and connexin hemichannel opening in disrupting normal homeostasis [30], we investigated whether blocking pannexin channel opening would enhance the previously described therapeutic effect of connexin hemichannel block.

One reported pannexin channel blocker is probenecid. It is used in the treatment of chronic gout or gouty arthritis caused by increased quantities of uric acid in the blood [31,32]. In the treatment of gout, inhibition of Panx1 channels did not affect connexin hemichannels [31]. Probenecid inhibits Panx1 channel currents in a concentration-dependent manner and by interacting with the first extracellular loop of Panx1 [31,33]. In the plasma, the t_1/2_ of probenecid varies from 5 h to 8 h.

A specific connexin hemichannel blocker is the novel benzopyran compound (–)-*cis*-6-acetyl-4*S*-(3-chloro-4-fluorobenzoylamino)-3,4-dihydro-2,2-dimethyl-2*H*-benzo[b]pyran-3*S*-ol (SB-220453), also known as tonabersat [25,34]. Our previous studies have shown that tonabersat confers protection to the injured retina at concentrations ranging between 10–90 μM [13,15,20]. Tonabersat at doses of 0.1–30 μM directly reduces opening of hemichannels under pathological conditions [13,15] while gap junctions may be affected at doses >100 μM in vitro. However, the drug has established human pharmacodynamic and pharmacokinetic parameters and human safety in Phase 2 trials (given to >1500 patients for up to 20 weeks once daily). Two-year carcinogenicity and high dose toxicity studies have also been completed. It can be administered by systemic or oral routes (oral t_max_ absorption of 0.5–3 h; plasma t_1/2_ = ~40 h with no accumulation) and is safe in humans [35]. Despite previous in vitro reports, it does not seem tenable that it is uncoupling gap junctions in physiological in vivo conditions [15,36,37].

In this study, novel data on the effect of selective inhibition of connexin hemichannel and pannexin channel opening, independently and together, were compared and discussed using existing data on the therapeutic effect of tonabersat and other connexin hemichannel blockers. The aim is to determine the effect of pannexin channel and connexin hemichannel blockers on retinal function in the light-damaged rat eye.

## 2. Results

### 2.1. Effects of Probenecid and/or Tonabersat on the Electroretinogram (ERG)

The a-wave of the ERG is the combined electrical activity of rods and cones that can be directly measured, to reflect function of the photoreceptors. Similarly, the b-wave is the positive deflection in the ERG waveform and is an objective measure of the function of the inner retina. In the light damaged retina, abnormal function of these cells precedes morphological damage and is a reliable quantitative marker for assessment of the therapeutic effect. Bright light damage leads to an almost total loss of retinal function within 24 h and no subsequent significant recovery [38]. However, we have shown that systemic delivery of tonabersat alone (by double intraperitoneal injection, one during light damage and one after cessation of light exposure) significantly improved retinal function assessed by ERG 1 week after treatment, and was significantly better again at 2 weeks post treatment [15]. Similarly, we previously have shown that oral tonabersat (10–90 μM) delivered during light damage significantly improved function at 1 week, 2 weeks and 3 months post-treatment [13] compared to retinas from vehicle-only treated animals. Saline treated animals have severely impaired retinal function after light damage and show no recovery over time.

To determine whether blocking other ATP-releasing channels also improves retinal function, we tested the effect of blocking pannexin channels alone, or block of pannexin channels in combination with the connexin hemichannel blocker tonabersat, comparing the drug effect on ERG profiles 24 h, 1 week and 2 weeks post intense light exposure.

The saline-injected rats, as usual for this model, did not recover after light-induced damage, with reduced function at all time points (Figure 1). At 24 h post treatment, there were no significant differences between the 1 mM probenecid, 1 mM probenecid with 30 µM tonabersat and the saline treatment groups (Figure 1). The ERG amplitudes for the drug-treated rats at 1 week and 2 weeks were, however, different from saline (Figure 1a–c) indicating recovery in the drug-treated groups. The rod and cone mixed responses were recorded after the initial flash, and the response from the second flash was recorded and represents the function from the cones only. The rod PIII response was derived through digital subtraction of the cone response from the initial mixed response. The a-wave amplitude in the probenecid and tonabersat combination treatment at 1 week was significantly different from saline at stimulus 1.1 log cd.s/m^2^ (ANOVA, *p* < 0.05) and 1.6–2.1 log cd.s/m^2^ (*p* < 0.01) (Figure 1d). At 2 weeks post light damage, intensities 0.1–2.1 log cd.s/m^2^ in the probenecid and tonabersat combination group resulted in a higher a-wave amplitude when compared with saline injected rats (*p* < 0.001), but the amplitude was significantly lower than normal a-wave values (Figure 1d).

Rats treated with probenecid alone also recovered from intense light damage starting 1-week post treatment, where the mixed a-wave amplitude of the ERG was increased significantly and with similar values to the combination treatment (Figure 1e). After 2 weeks, the mixed a-wave amplitude was larger and significantly different from the saline group at intensities 0.1 log cd·s/m^2^ and higher (*p* < 0.001) (Figure 1e). The a-wave amplitude of both treatment groups was similar at 1 week, although there was a 100 µV improved a-wave function at 2 weeks post-injury in the combination group compared to probenecid alone.

Further analysis showed that the mixed b-wave amplitude of the ERG also improved as a function of time and drug-treatment. While the ERG b-wave amplitude of the probenecid with tonabersat treatment group did not improve until week 2 (Figure 1f), the mixed b-wave amplitude in the probenecid only group showed a significant increase starting at 1 week post-treatment at light intensities of 0.1 to 2.1 log cd·s/m^2^ (*p* < 0.05) (Figure 1g). However, at 2 weeks both treatment groups showed a similar recovery in the b-wave amplitude (*p* > 0.05) (Figure 1f,g).

### 2.2. Effects of Probenecid and/or Tonabersat on the PIII, PII and OPs of the ERG

The rod and cone PII (the bipolar cell component) and rod PIII (the photoreceptor component) of the ERG can be isolated from the mixed a-wave and b-wave responses. The oscillatory potentials (OPs) correspond to the function of inner retinal cells including amacrine cells. Breakdown analysis of isolated rod PIII amplitudes after probenecid with tonabersat combination treatment, extracted at the highest flash intensity of 2.1 log cd.s/m^2^, showed progressive recovery of the rod photoreceptor activity through to the 2-week time point compared to the saline-treated group (*p* < 0.001; Figure 2a). A similarly improved rod PIII amplitude was observed after probenecid-only treatment (Figure 2b). Analysis of the rod post-photoreceptor pathway (rod PII amplitude) extracted from the b-wave response showed significant changes at 2 weeks from rats treated with probenecid and tonabersat (*p* < 0.001; Figure 2c). Similarly, the rod PII amplitude in probenecid only treated rats was significantly different at 2 weeks compared to saline (*p* < 0.05). Cone PII amplitude indicative of the response in the post-photoreceptoral pathway was significantly greater at 2 weeks post light damage in the probenecid with tonabersat group (p < 0.001, Figure 2e) and in the probenecid alone group (*p* < 0.05, Figure 2f). The effect of treatments on the OP amplitude of the ERG was significantly different with saline and between treatments. Tonabersat with probenecid treatment significantly improved the amplitude of OPs at 1 week and 2 weeks (Figure 2g). Similarly, probenecid treatment significantly improved OP amplitude recovery at 1 week and 2 weeks (Figure 2h). Overall, however, the combination of tonabersat with probenecid showed lower recovery whilst probenecid resulted in a greater and faster recovery of OPs amplitude beginning at 1 week.

### 2.3. Effect of Probenecid Alone or Tonabersat with Probenecid on Retinal and Choroidal Thickness

Bright light damage leads to retinal and choroidal degeneration over time, continuing weeks and even months after the insult [13,15]. The effect of the two treatments on protection against structural degeneration in both the retina and choroid was analyzed using optical coherence tomography (OCT). It can be seen from the data in Figure 3 that there was thinning of retinal layers and choroid in the light-damaged saline treated group 2 weeks post intervention. In comparison, the retina in the probenecid and probenecid with tonabersat treatment groups was preserved and not significantly different from baseline non-light damaged rats. Further analysis of outer nuclear layer (ONL) thickness showed that either probenecid alone or probenecid with tonabersat preserved ONL thickness at all time points (*p* < 0.001; Figure 3a,b). Moreover, the effect of both treatments protected the choroid from thinning and this effect was maintained and significantly different from the saline treated group (*p* < 0.001; Figure 3c,d).

### 2.4. Treatment Effect on Expression of Cx43, Retinal Gliosis and Inflammation

Following assessment of tissue structure, the effect of tonabersat with probenecid combination, probenecid alone or saline on key retinal inflammatory markers was investigated using several tissue markers. Markers included Cx43, the target of the connexin hemichannel blocker, tonabersat. Cx43 is commonly upregulated under injury conditions although it is difficult to differentiate between gap junctions and hemichannels, and it is not possible to distinguish function based upon labelling alone. The other markers were ionized calcium-binding adapter molecule (Iba-1) to evaluate microglia immunoreactivity pre- and post-treatment, and glial fibrillary acidic protein (GFAP) to investigate the extent of astrogliosis and Müller cell activation.

As can be seen in Figure 4, high Cx43 immunoreactivity was observed in the retina in the saline group of light-damaged eyes (Figure 4a). This is higher than in the normal Sprague Dawley (SD) rat retina [12] and was also highly increased in astrocytes surrounding blood vessels. Treatment with probenecid and/or tonabersat with probenecid caused a reduced immunoreactivity of Cx43 in the retina compared to the saline group (Figure 4b,c).

The Iba-1 antibody was used to analyze microglial activation. The protective effect of probenecid and/or and tonabersat and probenecid was assessed by counting the total number of Iba-1 positive microglia cells present in the tissue compared with light-damaged rats. There were fewer Iba-1 labelled cells in the inner plexiform layer and reduced evidence of migration of microglia to the outer layer of the retina (Figure 4e,f) in both treatment groups compared to the saline group (Figure 4d).

Finally, GFAP was used to determine the effect of probenecid and/or and tonabersat with probenecid on the extent of astrocytosis and Müller cell activation, particularly in the inner retina. Intense GFAP immunoreactivity associated with ganglion cell layer astrocytosis and with expression of GFAP in the processes of Müller cells was detected in the saline group (Figure 4g). Based on the results, there was less immunoreactivity for GFAP in both probenecid plus tonabersat and probenecid alone treated rats (Figure 4h,i) compared to the saline group (Figure 4g).

Quantification of the immunohistochemistry labeling revealed significantly lower levels of Cx43 expression in the tonabersat with probenecid treated group compared with saline as well as in the probenecid alone treated group compared with saline where Cx43 was significantly upregulated (*p* < 0.001) (Figure 5a). Quantification of the Iba-1 positive cells revealed that treatment reduced the number of active microglia compared with saline in both treatment groups, probenecid with tonabersat combination and probenecid alone (*p* < 0.001) (Figure 5b). However, the tonabersat and probenecid combination group also had statistically significantly fewer microglia compared to the probenecid group *(p* < 0.05) (Figure 5b). Similarly, the marker of retinal stress, expression of GFAP in Müller cells, showed consistent expression in the light-damaged, saline injected retina that was significantly reduced in both probenecid or probenecid with tonabersat treated groups (Figure 5c).

## 3. Discussion

The present study was conducted to evaluate the therapeutic effect of connexin hemichannel and pannexin channel modulators, tonabersat and probenecid, respectively, in the rat bright light-damaged retina model, a model that mimics aspects of inflammation in degenerative retinal disease. In general, the results suggest that there is a strong association between functional and structural recovery of the light-damaged retina when treated with probenecid alone and in combination with tonabersat but with significant differences in functional recovery by retinal cell type. There were differences in the site of retinal action (outer versus inner retina) and choroid. The effects were also different as a function of time.

The combination of connexin and pannexin modulators effectively reduced inflammation markers up to 2 weeks post light damage. These results suggest that blocking connexin hemichannels is needed and is in agreement with our previous study demonstrating that tonabersat alone was able to improve retinal function and preserved the retinal layers in the light-damaged rats [13,15]. The results also open discussion of the effect of probenecid on pannexin channels and connexin hemichannels as—similar to brain injury studies [32,39]—a time-dependent drug effect was seen.

The ERG results obtained after probenecid treatment of the light-damaged rat retina mimics the ERG results obtained in Panx1^−/−^ mice, where the a-wave and b-wave amplitudes were increased compared to the Panx1^+^ control [40]. This suggests that pannexin channels have a role in temporal modulation of light signal transmission and that protection conferred in the light-damaged rat is compatible with pannexin channel location on bipolar cells. We have shown that blocking pannexins results in the enhanced rod post-photoreceptor pathway involving bipolar cell and amacrine cell function. In fact, several studies have revealed that through release of ATP via the Panx1 channel, paracrine functions are enhanced including release of further signals for apoptotic cell clearance [41,42,43]. However, a significant role of pannexins in tissue homeostasis other than in cell death is more likely [11,30]. In the adult mouse retina, pannexin labeling is observed in retinal ganglion cells, horizontal cells, intensively at the axon terminal of subtypes of OFF bipolar cells and at the contact between rods and cone bipolar cells [11]. Blocking the pannexin channels after light exposure, similar to the results in the Panx1^−/−^ model may not be related to inflammation, but to action on signal transmission. Better recovery of ERG function was obtained with tonabersat [15] and no added benefit was seen when combining pannexin and connexin hemichannel block.

However, connexin hemichannel blockers delivered by other routes seem to have a more substantial effect on this rat model (Table 1). For example, Peptide5 after i.v. injection 2 h into [16] or injected twice, at 2 h and at the end of light damage period [14] or orally, once-only fed to rats immediately before the light damage period [13].

The connexin hemichannel blockers compared with probenecid were Peptide5 and tonabersat. Peptide5 is a mimetic peptide matching a portion of the extracellular loop 2 of the Cx43 protein and acting upon the same loop, adjacent to the region from which it is derived [44]. It has shown efficacy as a hemichannel blocker in multiple animal models (see for example [5]). It is likely to be Cx43 specific but has high homology with some other alpha group connexin isoforms (for example Cx40). Tonabersat is a novel *cis*-benzopyran derivative that has been shown to be equivalent to Peptide5 as a hemichannel blocker (for example cf. [13] with [14] and this manuscript). It is not connexin isoform specific (unpublished data). The effects of connexin hemichannel and pannexin channel block were not additive. This non-additive effect has similarities with an in vitro ischemia/reperfusion study, where about a third of the ATP release was pannexin mediated during the ischemia phase, two thirds was through hemichannels, but post-ischemia (reperfusion phase) only connexin hemichannels were responsible for ATP release [15]. Furthermore, in a study of ischemia/reperfusion, probenecid with Peptide5, and probenecid alone had less effect at reducing ATP release compared with Peptide5 alone [45].

Previous studies on the effect of connexin inhibition in the light-damaged rat retina are limited to those conducted by our group (Table 1). To determine the scale of tonabersat and probenecid effect we compared the outcomes of this current study with the connexin hemichannel blocker tonabersat also delivered by i.p. injection twice, at 2 h and at the end of light damage period in the bright light-damaged retina model [15], with peptide delivered using sustained nanoparticle delivery [14,16], and orally delivered tonabersat [13]. In doing so, we are comparing acute delivery of peptide (with a short half-life) with sustained delivery (in nanoparticles—important in the comparison with tonabersat which has a long half-life), we are also comparing intraperitoneal delivery versus oral delivery of tonabersat, and we are comparing tonabersat with peptide5, and finally we are showing consistency (reproducibility) with connexin hemichannel blockers across multiple studies.

The effect on the ERG from blocking connexin hemichannels, pannexin channels, or both, was visualized in a comparative graph. In general, the a-wave of the ERG reflects the rod and cone photoreceptor response in the outer retina, while the post-photoreceptor response of cells in the inner retina, mainly bipolar cells, is reflected by the b-wave. The normal ERG values for SD rats ranged between −400 to −700 µV for the a-wave, and between 400 to 700 µV for the b-wave at the highest light intensity [46]. The most obvious finding that emerged from the comparative analysis is that treatment with connexin modulators conferred similar improvement in retinal function of the outer retina compared with the pannexin channel modulator. The combination of pannexin and connexin hemichannel block was not additive. Specifically, the a-wave amplitude in the ERGs improved with all treatments, and values were within the normal range for most of the Cx43-based treatments trialed (Figure 6a). Probenecid alone or in combination significantly ameliorated the response, but not to a normal level (Figure 6a). The mixed b-wave amplitude of the ERG improved with all treatments, but probenecid was below normal values, tonabersat alone conferred better recovery and improvements to a normal range when using probenecid was only seen with the combination of drugs (Figure 6b).

Analysis of the rod pathways, reflected in the amplitude of the rod PIII (phototransduction of rods) and rod PII amplitude (rod bipolar cells) showed almost every treatment improving rod pathway function to a normal range (Figure 7a) except probenecid or tonabersat alone delivered by intraperitoneal injection. The recovery effect varied depending on the formulation. The comparative analysis of the rod PII amplitude, on the other hand, showed that all pannexin channel or connexin hemichannel blockers had a positive effect on the recovery of the light-damaged rat retina (Figure 7b). Oral delivery and longer treatment had an additional effect. Combination treatment was not additive.

Analysis of the cone PII amplitude, indicative of the cone bipolar cell response showed that inner retinal activity was ameliorated to a normal range using any of the formulations (Figure 8a). Similarly, the specific function of amacrine cells, revealed by the OPs amplitude analysis showed that better recovery of function was obtained by each formulation alone than by a combination of tonabersat and probenecid (Figure 8b).

So far, the Panx1 channel has been identified as being a potentially important part of P2X receptor activation through ATP release and by eliciting a pannexin mediated immune response. However, specific action of connexin hemichannel blockers better improved the function of the retina [14], thus demonstrating that in vivo as well as in vitro specific modulation of connexin hemichannels can confer neuroprotection [47], overriding any effect of blocking pannexin channel-mediated ATP release. Even though probenecid alone improved retinal function and morphology, protection was more pronounced by tonabersat and other connexin hemichannel blockers [13,15,16]. We also identified a role for microglia in the light damage model. During cellular insults, reactive microglia migrate to the outer retina [48,49]. According to Rutar et al. [50], inflammation is present in both the retina and the choroid following intense light damage suggesting residential microglia/macrophage activation play a major role in the process of tissue damage in the light-damaged albino rat model. The connexin hemichannel mediator role in pathology was shown by the number of microglia which was significantly lower in the probenecid plus tonabersat treated group compared to probenecid treatment alone, possibly owing to connexin hemichannels remaining open. The findings in our study also indicate that GFAP immunoreactivity and Cx43 immunoreactivity in the tonabersat with probenecid group was ameliorated. These findings are in parallel with a previous study using the Peptide5.

## 4. Materials and Methods

### 4.1. Light Damage Procedure

All experimental procedures were approved by the University of Auckland Animal Ethics Committee and comply with the Association for Research in Vision and Ophthalmology (ARVO) statement for the use of animals in eye research. Six- to eight-week old Sprague-Dawley (SD) rats (200–250 g; male or female) were used in these experiments. In total, 18 rats were used to create a model of retinal degeneration induced by intense light exposure and animals were divided into three experimental groups (control: saline, treatment: probenecid or probenecid with tonabersat). The control and treatment groups were randomized for statistical analysis. The analysis included comparison with six normal uninjured six to eight weeks-old SD rats (200–250 g; male or female) maintained under similar conditions to the treatment groups.

Light damage was induced by exposure to fluorescent light tubes (Master TLD 18W/965, Philips, Eindhoven, The Netherlands) with 2700 lux luminance when placed directly above the rat cages. The light has an emission range of 380 to 760 nm; no heat is generated by the fluorescent light. Animals were allowed to roam free in the cage during the 24 h light exposure and had free access to food and water. Light damage experiments were performed consistently around 9:00 am. After light exposure, the animals were returned to normal light-dark cycle conditions (12 h light at 174 lux and 12 h darkness at <62 lux) to recover for 24 h; eyes were tested and returned to these light conditions for 1 week. Tissues were collected after the final eye tests at 2 weeks.

### 4.2. Animal Drug Administration and Anaesthesia Procedures

Treatment consisted of a final predicted blood circulating concentration of 1 mM probenecid, or a combination of both 1 mM probenecid and 30 µM tonabersat (prepared as outlined below in 2.3); the control consisted of a similar volume of 0.9% NaCl (saline) injection. Animals were treated twice by an intraperitoneal (i.p.) injection 2 h after the onset and immediately after the completion of the 24 h light exposure. All i.p. injections were conducted using a syringe attached to a 27 G x ½ inch needle (BD PrecisionGlide™, Becton-Dickinson and Co., Franklin Lakes, NJ, USA). Tonabersat and probenecid solutions were i.p. injected consecutively, as combining both drugs while concentrated caused drug precipitation.

No anesthesia was applied during i.p. drug administration and animals were restrained by the over the shoulder grip technique [51]. All other manipulations were performed on anaesthetized rats using a combination of ketamine (75 mg/kg, Parnell Laboratories, Auckland, New Zealand) and domitor (0.5 mg/kg, Pfizer, Auckland, New Zealand) in saline. Following manipulations, anesthesia was reversed by i.p. injection of atipamezole (1 mg/kg antisedan, Pfizer) and the animals were returned to their cages and monitored during anesthesia recovery.

### 4.3. Tonabersat and Probenecid Preparation and Injection

The drugs were delivered via i.p. injection to allow comparison with the previous tonabersat injection protocol [15] and with known effective routes of probenecid delivery to the retina [52]. Tonabersat solution was prepared following a procedure previously described [15]. Briefly, tonabersat was weighed and added to polyethylene glycol (PEG) (molecular weight 400 g/mol; Sigma Aldrich, Auckland, New Zealand) and (2-hyrdoxypropyl)-β-cyclodextrin (Sigma Aldrich) vehicle mixture at a ratio of 10 mg per 10 mL PEG/cyclodextrin. The solution was mixed thoroughly and sonicated in a 35–40 °C water bath for 2 h. This drug solution was used fresh or stored at 4 °C for up to 2 weeks. Rats were injected with 0.24 mL of the 1 mg/mL tonabersat stock (equivalent to 1 mg/kg or 30 µM in blood), delivered 2 h into the light damage procedure and again at the end of the light damage procedure.

Probenecid (50 mg) was dissolved in 1 mL of 2 mM sodium hydroxide, and the pH was adjusted to 7.0 with 0.2 M hydrochloric acid [32]. Rats were treated with 0.115 mL of the probenecid 50 mg/mL solution (equivalent to 24 mg/kg) delivered 2 h into the light damage procedure and again at the end of the light damage procedure. Figure 9 shows the experimental set up and outline of the intervention protocols.

### 4.4. Electroretinogram

The procedure was performed as described previously [12,53]. SD rats were dark-adapted overnight for 12–14 h before the electroretinogram (ERG) recordings. The ERG was recorded before light damage (baseline), and after recovery from treatment at 24 h, 1 week and 2 weeks. A dim red light generated by a light-emitting diode (λ_max_ = 650 nm) was used during manipulation of dark-adapted animals. The corneas were maintained hydrated with 1% carboxymethylcellulose sodium (Celluvisc, Allergan, Sydney, Australia) throughout the entire ERG recording procedure using gold ring electrodes (Roland Consult Stasche & Finger GmbH, Brandenburg, Germany). The active electrode was U-shaped and was kept in contact with the centre of the cornea. The inactive electrode was V-shaped and hooked around the front teeth and in contact with the wet tongue. Body temperature was maintained at approximately at 37 °C using a warm heating pad to avoid temperature-driven ERG amplitude fluctuation.

Full-field ERG responses were elicited by a twin-flash (0.8 ms second stimulus interval) generated from a photographic flash unit (SB900 flash, Nikon, Tokyo, Japan), via a Ganzfeld sphere. In the Ganzfeld method, an integrating sphere approximately 650 mm in diameter, painted white internally was used to reflect light to the pupil and illuminate the entire retina. The flash intensity range was from −2.9 to 2.1 log cd.s/m^2^ and was attenuated using neutral density filters (Kodak Wratten, Eastman Kodak, Rochester, NY, USA), to obtain light intensities of −3.9, −2.9, −1.9, 0.1, 1.1, 1.6, 1.8 and 2.1 log cd.s/m^2^. The flash intensity was calibrated using an IL1700 research radiometer (UV Process Supply Inc., Chicago, IL, USA). This study utilized a twin-flash paradigm for the isolation of rod and cone pathways. Paired flashes of identical luminous energy were triggered from the flash unit. The rod and cone mixed responses were elicited after the initial flash, and the cones-only response was obtained after the second flash at high light intensities. The rod photocurrent (rod PIII response) was derived through digital subtraction of the highest light cone response from the corresponding initial mixed response [53]. The bipolar cell component of the ERG response is the rod or cone PII obtained also through computational analysis as previously described [14]. The oscillatory potentials (OPs) summed amplitude were isolated by subtracting the initial b-wave response from the rod PII [54]. Recordings were performed in a Faraday cage to reduce electrical noise. The results of ERG signals were amplified 1000 times by a Dual Bio Amp (AD Instruments, NSW, Australia) and waveforms were recorded by using the Scope software (AD Instruments) and analyzed using published algorithms of the amplitudes of a-wave and b-wave of the ERG [14,46,53].

### 4.5. Optical Coherence Tomography

The optical coherence tomography (OCT) approach was used to obtain information on in vivo retinal layer morphology and any changes induced by light damage. A spectral domain optical coherence tomography (SD-OCT; Micron IV; Phoenix Research Labs, Pleasanton, CA, USA) was employed. This procedure was executed immediately after ERG recordings under anesthesia and pupil dilation [14]. OCT was recorded before light damage (baseline), and after each ERG at 24 h, 1 week and 2 weeks. Rats were placed on a warm heating pad to maintain normal body temperature and to prevent the development of cold cataracts. Dilated eyes were covered with Poly Gel (containing 3 mg/g Carbomer; Alcon Laboratories, Geneva, Switzerland), and the retina was imaged by contacting the OCT lens to the gel. The superior-central region of the retina was used in the imaging procedure since it has been found to be the most affected area after bright light exposure [55]. StreamPix 6 software, version 7.2.4.2 (Phoenix Research Laboratories) was used for image acquisition. The SD-OCT horizontal line B-scan had 2 µm axial resolution, and consisted of 1024 pixels per A-scan. Ten B-scans acquired 2 mm from the optic nerve in the dorsal retina were taken and averaged. Images were analyzed using InSight software, version 1.1.5207 (Phoenix Research Laboratories) to calculate the thickness of retinal and choroidal layers. The choroidal layers were measured from the hyper-reflective Bruch’s membrane to the choroidal-scleral interface. The outer nuclear layer was also was measured from the RPE hyper-reflective line to the outer plexiform layer. To avoid overestimation of retinal thickness, all measurements were taken using caliper lines drawn perpendicular to the retina on a vertical image.

### 4.6. Tissue Collection, Processing and Immunohistochemical Labelling

At the end of the final ERG and OCT procedure (week 2), rats were immediately culled and tissue collected. Rats were perfused transcardially with saline for 2–3 min followed by 30 min perfusion with 4% paraformaldehyde (PFA) prepared in 0.1 M phosphate buffer (PB). The eyes were dissected from the orbit, the anterior segment removed and eyecups were further fixed in 4% PFA for 30 min followed by washes in 0.1 M phosphate buffer saline (PBS), pH 7.4. After processing in increased sucrose solutions up to 30% sucrose in PBS, tissues were submerged in optimal cutting temperature medium (OCT, Sakura Finetek, Torrance, CA, USA) and frozen for sectioning. Cryo-sections were 12 μm in thickness and were collected on Super-frost Plus slides (Labserv, Auckland, New Zealand).

Imunohistochemical labelling was conducted using the indirect immunofluorescence technique. Briefly, sections were washed with 0.1 M PB and blocked with a solution containing 6 % normal goat serum or donkey serum (Invitrogen, Grand Island, NY, USA), 1% bovine serum albumin (BSA) and 0.5% Triton X-100 in 0.1 M PBS for 1 h at room temperature. Afterwards, tissues were immunolabelled with primary antibodies, including rabbit anti-Cx43 (1:1000, Cat C6219, Sigma Aldrich), mouse anti-ionized calcium-binding adapter molecule (Iba-1, 1:250, Cat Ab5076, Abcam, Cambridge, MA, USA) and mouse anti-glial fibrillary acidic protein (GFAP, 1:1000, Cat C9205, Sigma-Aldrich). Antibodies were diluted in 3% normal goat serum, 1% BSA and 0.5% Triton X-100 in 0.1 M PBS. Tissue sections were incubated with the primary antibody overnight at room temperature. After the incubation period, the sections were washed four times for 15 min each in 0.1 M PB in order to remove the excess of primary antibodies. The secondary antibody, goat anti-rabbit or goat anti-mouse, were conjugated with Alexa TM 488 or Alexa TM 594 (Invitrogen) diluted 1:500 in the primary antibody buffer and applied for 2–3 h in the dark, at room temperature. The slides were then washed thoroughly with 0.1 M PB to remove excess secondary antibodies. Slides were mounted in anti-fade medium (Citifluor Ltd., Leicester, UK) and sealed with nail polish. A confocal laser scanning microscope, Olympus FluoView FV1000 (Olympus Corporation, Tokyo, Japan), fitted with 405, 473 and 559 nm excitation wavelength solid-state lasers was used. A series of four to eight images per eye in each experimental group were used in the image analysis. A z-series of optical sections were collected at 1 µm step-size perpendicular to the optical axis (the z-axis). Sequential image acquisition was automatized maintaining the same parameters for all tissues as previously described [12,14]. Images were obtained from a region of interest, using a 20 X UPLSAPO NA:0.75 objective with pinhole 1 AU, λ_EX_ = 473 nm or 559 nm. Images size were 1024 pixels over 635 µm (resolution of 0.62 µm per pixel). The images are presented as an intensity projection over the Z axis, a standard feature of the image acquisition and processing software (Olympus Fluoview ver4.2b). The images were collected from a 4–8 µm depth region to represent labelling falling on one cell average (retinal cell sizes varies from 7–20 µm). Six retinas obtained from different animals were analyzed in each group. Data were reported as number of pixels corresponding with the GFAP or Cx43 antibody labelling/unit area. The mean number of Iba-1 positive cells was counted in selected areas. Microglia were identified by their ramified network of processes originating from the cell soma. ImageJ software (National Institutes of Health, Bethesda, MA, USA) and appropriate plugins (bandpass filter, unsharp mask) were consistently used to identify Iba-1 positive cells. The number of Iba-1 cells were averaged from six retinas per group.

### 4.7. Statistical Analysis

Graphing and statistical analyses were performed using GraphPad Prism 5 (GraphPad Software, La Jolla, CA, USA). All data is presented as the mean ± the standard error of the mean (SEM). Functional and morphological data (at least *n* = 6 per group) were compared using analysis-of-variance with an alpha value of 0.05. Data from this study and previous publications (Table 1) was compared using one-way ANOVA. A two-way ANOVA followed by a Bonferroni post-test was used in the ERG response analysis to compare the effects of stimulus intensity (recovery time vs. treatment groups). One-way ANOVA followed by Tukey’s test was used in the analysis of the ERG response at the highest light levels. Statistical analysis of rod PII and PIII was performed using an unpaired t-test with a Welch’s correction, assuming the distribution of means across samples was normal as it was in our previous studies [13,14,16].

## 5. Conclusions

These findings suggest that intraperitoneal injection of either tonabersat with probenecid or probenecid alone protects at least in part against the sequence of cellular events associated with retinal degeneration that results from intense light damage. Whilst both pannexin channel and connexin hemichannel block are effective, they are not additive suggesting that one may precede the other, consistent with previous results suggesting pannexin channels may open initially, but are self-regulating [8], and that it is connexin hemichannel opening that is longer lasting and contributes to chronic disease [15,16]. However, the effects were not identical either, reflecting the specific locations of the respective channels in different retinal cell types, and with pannexins block showing its biggest effect in protecting bipolar cell and amacrine cell function. Overall, however, comparison across studies shows that connexin hemichannel block alone is the most effective, almost entirely protecting retinal function on its own, with a single oral dose of tonabersat the most effective all treatments.

## 6. Patents

PCT/US20/50579 Green CR, Mat Nor MN, Acosta ML, Duft BJ. Compositions and methods for rescuing retinal and choroidal structure and function. US 62/900,379 & US 62/903,504 Green CR, Mat Nor MN, Acosta ML. Connexin hemichannel block using orally delivered tonabersat improves outcomes in animal models of retinal disease.

## Figures and Tables

**Figure 1 ijms-22-01755-f001:**
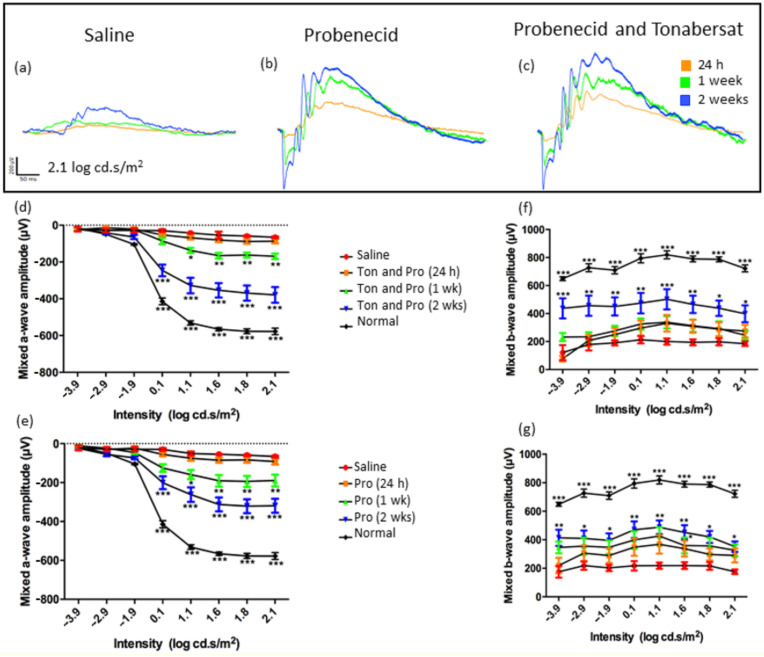
Differential effect of pannexin channel inhibitor (1 mM probenecid) in combination with connexin hemichannel block (30 μM tonabersat) and pannexin channel block alone (1 mM) on ERG amplitude. Representative ERG amplitudes obtained at 24 h, 1 week and 2 weeks post-treatment or saline (trace shown at 2 weeks post-injury) intraperitoneal injection in the intense light-exposed rat (**a**–**c**). The a-wave amplitude of the ERG in rats treated with a combination of tonabersat and probenecid (**d**) or probenecid alone (**e**). The b-wave amplitude of the ERG is shown for the tonabersat with probenecid (**f**) or probenecid-only treated rats (**g**). The graphs include the results from saline injection in the light-damaged rats (red; shown here at the 2-week time point) and the ERG response in normal uninjured rats (black). Saline injection data and control animals’ responses are plotted in all graphs. The mixed a-wave amplitude was significantly larger in the tonabersat and probenecid treated groups at selected intensities 2 weeks post-treatment compared with the probenecid group. The mixed b-wave amplitude was larger in the probenecid treated rats 1 week post treatment compared with the tonabersat with probenecid group. Statistical analysis was conducted using one-way ANOVA, followed by a post-hoc test. Ton = tonabersat, Pro = probenecid; * refers to significant values in comparison with saline-treatment: * *p* < 0.05; ** *p* < 0.01; *** *p* < 0.001.

**Figure 2 ijms-22-01755-f002:**
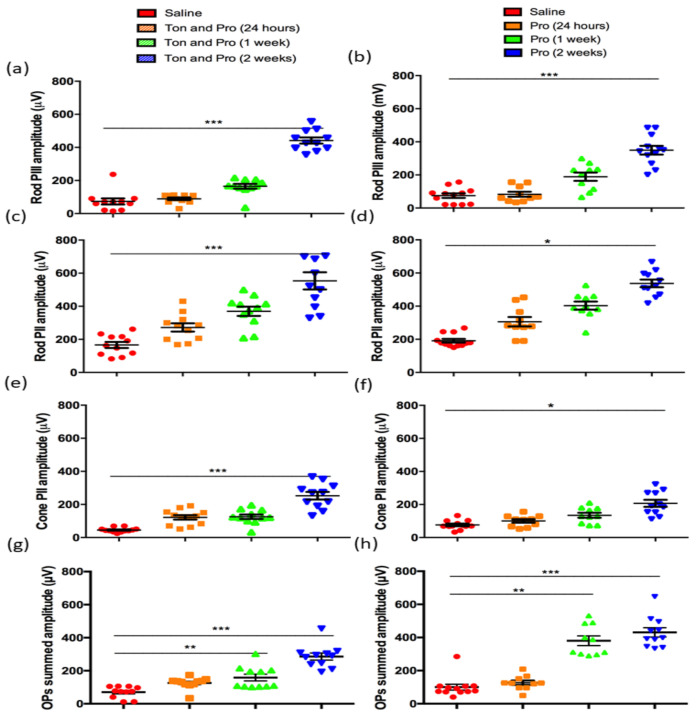
Representative rod PIII, rod PII and cone PII amplitudes of tonabersat with probenecid, probenecid and saline injected light-damaged rats. Rod PIII amplitude recovery is shown for the tonabersat with probenecid group (**a**) and probenecid alone (**b**). Rod PII amplitudes were also calculated for the tonabersat with probenecid group (**c**) and probenecid alone (**d**). The post-photoreceptor cone PII amplitude for the tonabersat with probenecid group is shown in (**e**) as is probenecid alone (**f**). The tonabersat and probenecid treated group showed a significantly recovered OP summed amplitude transiently at 1 week post-treatment (**g**) whereas the probenecid alone treated group showed greater improvement over time with significantly recovered OP summed amplitudes at 1 week and 2 weeks (**h**). Overall, both treatments resulted in the recovery of retinal function but probenecid showed a faster recovery of inner retinal function at 1 week and greater recovery of the OPs. Statistical analysis was conducted using one-way ANOVA, followed by post-hoc test. Abbreviations: Ton = tonabersat, Pro = probenecid; * refers to significant values in comparison with saline-treatment: * *p* < 0.05; ** *p* < 0.01; *** *p* < 0.001.

**Figure 3 ijms-22-01755-f003:**
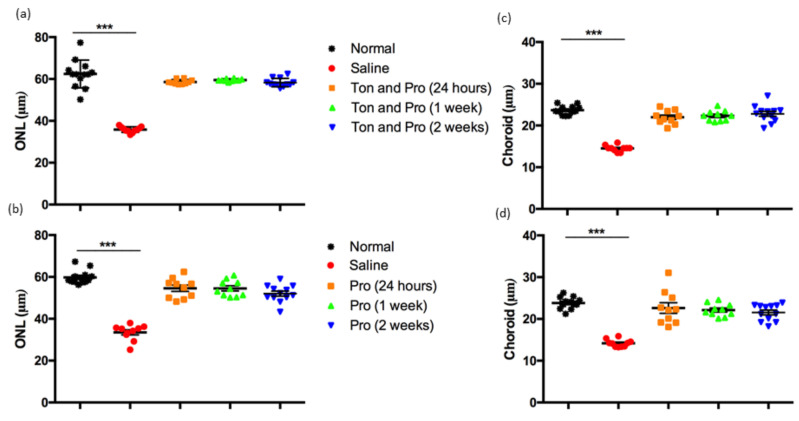
Effect of saline, probenecid with tonabersat and probenecid treatments on the light-damaged rat retinal and choroidal thickness. The ONL thickness was obtained from OCT images of tonabersat and probenecid treated rats (**a**), and the probenecid treated group (**b**). The choroidal thickness was measured by OCT analysis in the tonabersat and probenecid group (**c**) and probenecid group (**d**). Data are expressed as mean ± SEM. Abbreviations: Ton, tonabersat; Pro, probenecid; ONL, outer nuclear layer. Statistical analysis was conducted using one-way ANOVA, followed by a post-hoc test. Significant values are indicated with asterisks: *** *p* < 0.001.

**Figure 4 ijms-22-01755-f004:**
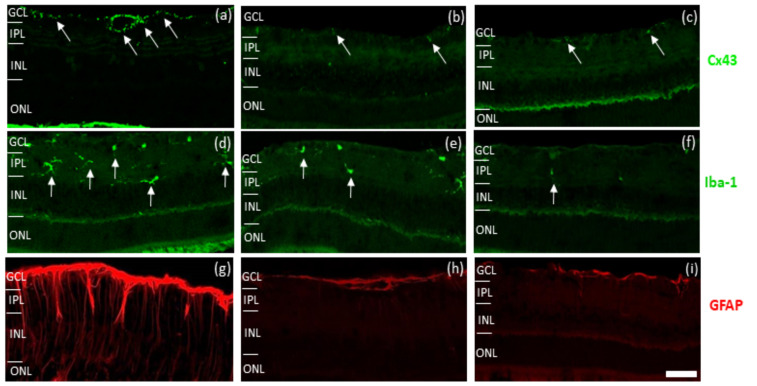
Effect of saline, probenecid and tonabersat combination and probenecid on immunoreactivity of light-damaged rats. Light-induced damage increased expression of Cx43 (green) in the ganglion cell layer (**a**). Probenecid and/or tonabersat treated rats showed less Cx43 immunoreactivity in the retina (**b**,**c**) compared to saline treated rats (**a**). Iba-1 immuno-labelled cells (green, highlighted by white arrows) were fewer in number in the retina of probenecid treated rats (**e**) and tonabersat and probenecid (**f**) compared to saline-treated rats (**d**). GFAP (red) immunoreactivity did not increase in the retina of probenecid treated rats (**h**) nor in the tonabersat and probenecid treatments (**i**) compared to saline-treated rats (**g**). Abbreviation: GCL, ganglion cell layer; IPL, inner plexiform layer; INL, inner nuclear layer; ONL, outer nuclear layer. Scale bar: 50 μm.

**Figure 5 ijms-22-01755-f005:**
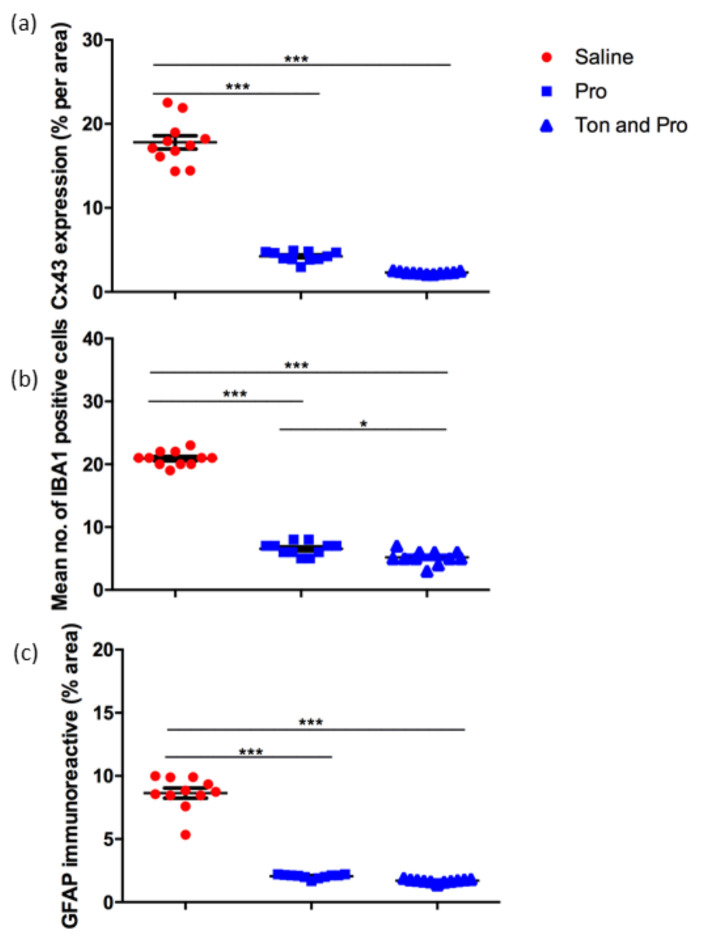
Quantification of Cx43 expression (**a**), mean number of Iba-1 positive cells/unit area (**b**) and GFAP immunoreactivity (**c**) in the saline, probenecid alone, tonabersat and probenecid combination treatment groups. Analysis revealed a significant decrease in Cx43, Iba-1 and GFAP in the treatment groups (**a**–**c**). The tonabersat and probenecid combination group further showed significantly fewer microglia compared to the probenecid group (**b**). Statistical analysis was conducted using one-way ANOVA, followed by Tukey’s multiple comparisons test. Significant values in comparison with results from the saline group are indicated with asterisks: **p* < 0.05; ****p* <0.001.

**Figure 6 ijms-22-01755-f006:**
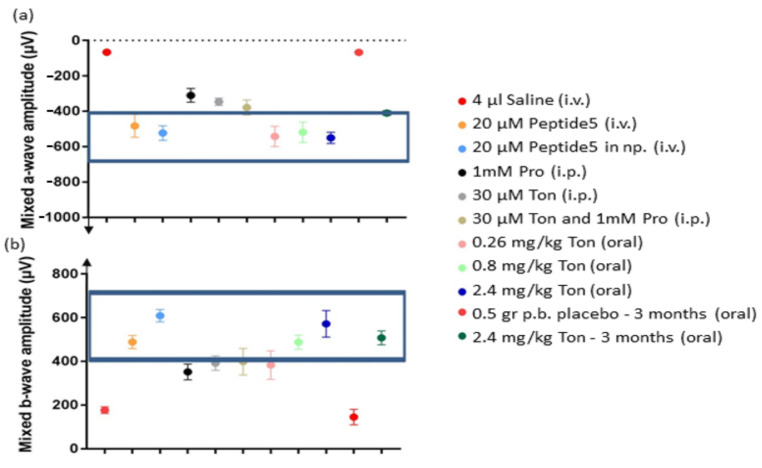
Overall analysis of the mixed a-wave amplitude which deflects downwards (**a**) and b-wave amplitude which deflects upwards (**b**) of the ERG for each of the connexin or pannexin blockers (or combination) applied to the light-damaged rat retina. The a-wave and b-wave amplitude in the ERGs improved towards the normal retinal function ranges for all the Cx43 hemichannel blocking treatments trialed compared to probenecid, which significantly ameliorated the response, but not to a normal level. Statistical analysis was conducted using one-way ANOVA, followed by a post-hoc test comparing computer-generated normal ERG values with treatment values. The normality values were randomly selected between −400 to −700 µV for the a-wave, and between 400 to 700 µV for the b-wave [46]. The boxed area shows treatments that preserved retinal function and were non-significantly different from normal values. Abbreviations: i.v., intravitreal; i.p. intraperitoneal; np, nanoparticles; Pro, probenecid; Ton, tonabersat; p.b. peanut butter.

**Figure 7 ijms-22-01755-f007:**
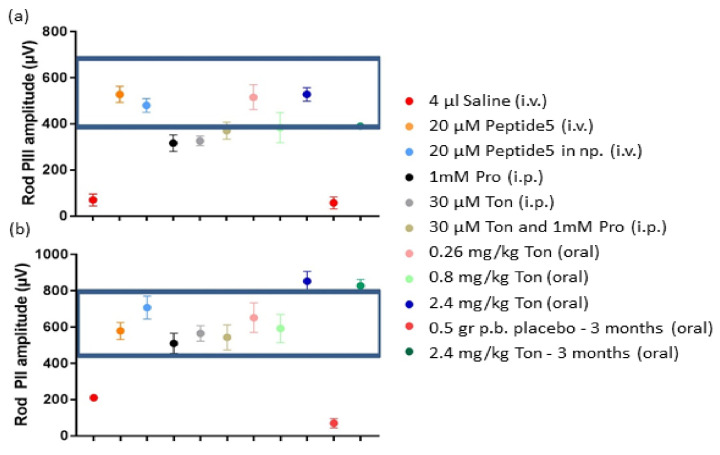
Overall analysis of rod PIII amplitude (**a**) and rod PII amplitude (**b**) for each of the connexin and pannexin treatments applied. In general, the rod photoreceptor response was better with tonabersat and Peptide5 treatments, and the lowest response was from the intervention using probenecid. The rod bipolar cell response (PII) was better with all treatments. Statistical analysis was conducted using one-way ANOVA, followed by a post-hoc test comparing computer-generated normal ERG values with treatment values. The normality values were randomly selected between 400 to 700 µV for the rod PIII amplitude, and between 400 to 800 µV for the rod PII amplitude. The boxed area shows treatments that preserved retinal function and were non-significantly different from normal values. Abbreviations as in Figure 6.

**Figure 8 ijms-22-01755-f008:**
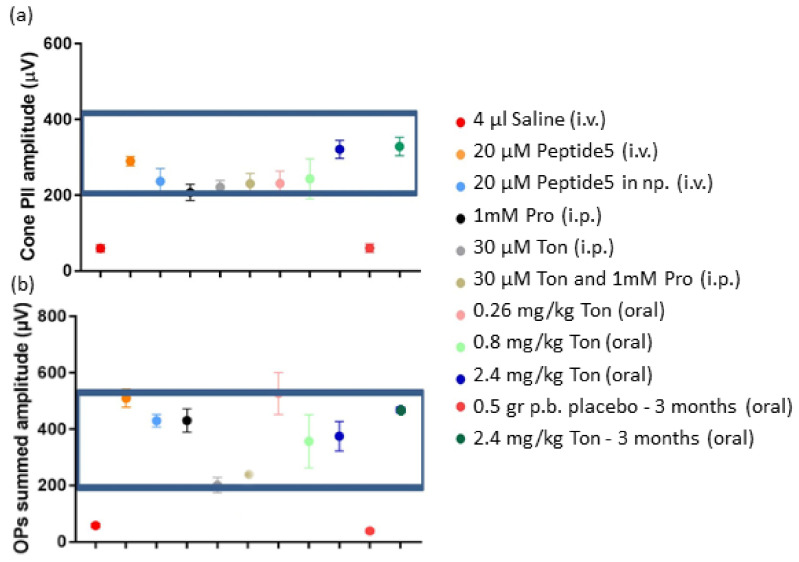
Analysis of cone PII amplitude (**a**) and summed OPs amplitude (**b**) for each of the treatment groups. Overall, the cone PII results indicate that the drugs used were able to protect the retina. However, the combination of tonabersat and probenecid or tonabersat alone delivered by intraperitoneal injection was less protective to the amacrine cells, which generate the OPs during signal transduction in the inner retina, than any of the other routes of delivery or probenecid alone. Statistical analysis was conducted using one-way ANOVA, followed by a post-hoc test comparing computer-generated normal ERG values with treatment values. The normality values were randomly selected between 200 to 400 µV for the cone PII amplitude, and between 200 to 500 µV for the summed OPs. The boxed area shows treatments that preserved retinal function and were non-significantly different from normal values. Abbreviations as in Figure 6.

**Figure 9 ijms-22-01755-f009:**
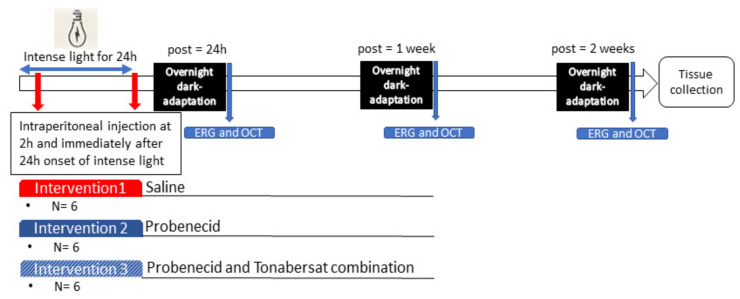
Diagram of the light exposure and intervention procedures for saline, probenecid and probenecid and tonabersat combination groups over a 2-week period. The intraperitoneal injections were performed 2 h after light-damage onset and immediately after 24 h of light exposure. Electroretinography (ERG) and optical coherence tomography (OCT) were performed at 24 h, 1 week and 2 weeks post light damage before enucleation of the eye.

**Table 1 ijms-22-01755-t001:** Summary of our published data reporting a therapeutic effect in the light-damaged rat model after intervention with Cx43 hemichannel blockers (tonabersat or Peptide5) with specific delivery methods.

Intervention	Delivery	Animals	Timeline	References
Control (vehicle)	Systemic (intraperitoneal and oral); ocular (intravitreal)	6	Follow up at 24 h, 1 week, 2 weeks, 3 months	[13,14,15,16]
Cx43 mimetic peptide (Peptide5)	Ocular (two intravitreal injections, one at 2 h and the other immediately after 24 h light exposure)	6	Follow up for 24 h	[14]
Peptide5	Ocular (one intravitreal injection)	6	Follow up at 24 h, 1 week and 2 weeks	[16]
Peptide5 in nanoparticles	Ocular (one intravitreal injection)	6	Follow up at 24 h, 1 week and 2 weeks	[16]
Tonabersat	Systemic (intraperitoneal injection at 2 h and immediately after 24 h light exposure)	6	Follow up at 24 h, 1 week and 2 weeks	[15]
Tonabersat	Systemic (oral administration immediately before starting light exposure)	6	Follow up at 24 h, 1 week, 2 weeks, 3 months	[13]

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
