# Peer review of "Differential Action of Connexin Hemichannel and Pannexin Channel Therapeutics for Potential Treatment of Retinal Diseases"

_ijms, 2021, doi:10.3390/ijms22041755_

Round 1

Reviewer 1 Report

The paper is well written with good scientific approach and results.

Author Response

Reply: We appreciate the reviewer’s comment and we thank you for your support.

Reviewer 2 Report

This is a brief but well-written study examining dysregulation of retinal function in early light-induced retinal degeneration and attempts to provide an explanation for differential actions of connexin or pannexin hemichannel block.  There are a few minor points to consider/change.

Line 61 when the author stipulates “Pannexin is expressed in the anterior part of the eye but in the retina”. Are they referring to a specific isoform? if so, this should be specified.

Lines 87 through to 93 should be a single paragraph and L94 indented to signify a new paragraph.

L97, can the authors please provide the range of concentration used in these studies i.e., REF 13 and 15.

L97 “Tonabersat directly reduces 97 opening of hemichannels under pathological conditions [13,15] without affecting gap 98 junctions” This statement infers that TB does not inhibit Gap Junctions, are the authors referring to a specific dose within their model? TB has been demonstrated to act as a GJ inhibitor and therefore the context in which they refer to TB as inhibiting hemichannels only needs to be described.

L103, please identify these earlier in vitro reports i.e., add appropriate references.

L112, 2.1 Please define ERG on 1st use in the subheading.  Check other abbreviations throughout, e.g., GFAP in L255, or OCT, etc.  Journal style has results ahead of methods and so abbreviations need to be defined in the results section to make the work more accessible.

Line 112. No detail is provided on the dose of Tonabersat or Probenecid in the results or legends, please amend to again make interpretation of results more accessible

L119-to-L125, in earlier studies, please can the authors mention if tonabersat was administered during the 24h period of bright light damage, or only after the damage had been caused. Protocol in the present study is 2 and 24h i.p. and matches the conclusion, which comments on protection.  However, have the authors examined recovery post damage as would more likely be the case for therapeutic intervention e.g., i.p. administration at 24h and beyond once the damage had been caused.

Line 377: Figures 6-8 lack any information regarding statistical significance between these data and this would help in interpretation of effect between the treatment groups

Author Response

Response to reviewer #2

1) This is a brief but well-written study examining dysregulation of retinal function in early light-induced retinal degeneration and attempts to provide an explanation for differential actions of connexin or pannexin hemichannel block.  There are a few minor points to consider/change.

Reply: Thank you for the suggested revisions of this manuscript. Please see our comments to your specific remarks below.

2) Line 61 when the author stipulates “Pannexin is expressed in the anterior part of the eye but in the retina”. Are they referring to a specific isoform? if so, this should be specified.

Reply: We agree with the reviewer. We had amended the sentence to indicate that Pax1 and Pax2 are expressed in the retina. Line 61: ‘Pannexin1 (Pax1) is expressed in the anterior part of the eye but in the retina Panx1 and Panx2 are widely distributed including ganglion cells and various cells of the inner retina [11].’

3) Lines 87 through to 93 should be a single paragraph and L94 indented to signify a new paragraph.

Reply: We have re-arranged the paragraphs as suggested by this reviewer.

4) L97, can the authors please provide the range of concentration used in these studies i.e., REF 13 and 15.

Reply: We have added the concentrations used at the end of the sentence in line 96:   ‘Our previous studies have shown that tonabersat confers protection to the injured retina at concentrations ranging 10- 90 mM [15,16,21] .’

5) L97 “Tonabersat directly reduces 97 opening of hemichannels under pathological conditions [13,15] without affecting gap 98 junctions” This statement infers that TB does not inhibit Gap Junctions, are the authors referring to a specific dose within their model? TB has been demonstrated to act as a GJ inhibitor and therefore the context in which they refer to TB as inhibiting hemichannels only needs to be described.

Reply: We agree. The dose-depended specific effect has been added to this statement. Line 97: ‘Tonabersat at doses 0.1-30 µM directly reduces opening of hemichannels under pathological conditions [13,15] while gap junctions may in vitro be affected at doses >100 µM. However……. ‘

Nevertheless, as noted, we know from previous preclinical toxicity and two year carcinogenicity studies that the drug does not uncouple gap junctions in vivo even at very high dose levels.

 6) L103, please identify these earlier in vitro reports i.e., add appropriate references.

Reply: The studies were conducted by Kim et al. We have added reference 15 (Kim et al) and Bialer et al, 2009, 2013 to this statement.

7) L112, 2.1 Please define ERG on 1st use in the subheading.  Check other abbreviations throughout, e.g., GFAP in L255, or OCT, etc.  Journal style has results ahead of methods and so abbreviations need to be defined in the results section to make the work more accessible.

Reply: The ERG abbreviation has been defined in the subheading. The manuscript has been checked and all abbreviations have been cited at first mention.

8) Line 112. No detail is provided on the dose of Tonabersat or Probenecid in the results or legends, please amend to again make interpretation of results more accessible.

Reply: The corresponding doses were added to the text (line 132) and Figure1 legend.

9) L119-to-L125, in earlier studies, please can the authors mention if tonabersat was administered during the 24h period of bright light damage, or only after the damage had been caused. Protocol in the present study is 2 and 24h i.p. and matches the conclusion, which comments on protection.  However, have the authors examined recovery post damage as would more likely be the case for therapeutic intervention e.g., i.p. administration at 24h and beyond once the damage had been caused.

Reply: Previous studies were conducted using intraperitoneal injections delivered during the light damage period and immediately after the cessation of light exposure. Orally-delivered tonabersat was administered once during the light damage period.  

Our unpublished results show that intervention in the light damaged retina 6 hrs and 24 hrs after the light-damage period using peptide 5 intraocular injections did not result in therapeutic benefit.

In a previous publication, we confirmed the acute and severe nature of light damage and the inability of any treatment to recover photoreceptor cells once they are already apoptotic (Yu et al., 2007).  However, the light damaged exposure time recreates the events seen in early stages of retinal damage (inflammation, channel opening, early stage cell death) and allow us to infer the therapy preventive effect.

TzuYing Yu, Monica L Acosta, Sarah Ready, YihLiang Cheong, Michael Kalloniatis. Light exposure causes functional changes in the retina: increased photoreceptor cation channel permeability, photoreceptor apoptosis, and altered retinal metabolic function. Journal of neurochemistry 2007; 103(2) 714-724

10) Line 377: Figures 6-8 lack any information regarding statistical significance between these data and this would help in interpretation of effect between the treatment groups

Treatment data was compared with computer-generated normal ERG values within ranges reported in SD rats [43].

Fig 6. Statistical analysis was conducted using one-way ANOVA, followed by a post-hoc test comparing computer-generated normal ERG values with treatment values. The normality values were randomly selected between -400 to -700 µV for the a-wave, and between 400 to 700 µV for the b-wave [43]. The boxed area shows treatments that preserved retinal function and were non-significantly different from normal values.

Fig 7 Statistical analysis was conducted using one-way ANOVA, followed by a post-hoc test comparing computer-generated normal ERG values with treatment values. The normality values were randomly selected between 400 to 700 µV for the rod PIII amplitude, and between 400 to 800 µV for the rod PII amplitude. The boxed area shows treatments that preserved retinal function and were non-significantly different from normal values.

Fig 8 Statistical analysis was conducted using one-way ANOVA, followed by a post-hoc test comparing computer-generated normal ERG values with treatment values. The normality values were randomly selected between 200 to 400 µV for the cone PII amplitude, and between 200 to 500 µV for the summed OPs. The boxed area shows treatments that preserved retinal function and were non-significantly different from normal values.